# Performance of triage systems in emergency care: a systematic review and meta-analysis

Joany M Zachariasse,[1] Vera van der Hagen,[1] Nienke Seiger,[1] Kevin Mackway-Jones,[2] Mirjam van Veen,[1,3] Henriette A Moll[1]

JMZ and VH contributed equally.

[1]Department of Pediatrics, Erasmus MC-Sophia Children's Hospital, Rotterdam, The Netherlands
[2]Emergency Department, Manchester Royal Infirmary, Manchester, UK
[3]Pediatrics, Groene Hart Ziekenhuis, Gouda, The Netherlands

**Correspondence to**
Dr Henriette A Moll;
h.a.moll@erasmusmc.nl

## ABSTRACT

**Objective** To assess and compare the performance of triage systems for identifying high and low-urgency patients in the emergency department (ED).

**Design** Systematic review and meta-analysis.

**Data sources** EMBASE, Medline OvidSP, Cochrane central, Web of science and CINAHL databases from 1980 to 2016 with the final update in December 2018.

**Eligibility criteria** Studies that evaluated an emergency medical triage system, assessed validity using any reference standard as proxy for true patient urgency and were written in English. Studies conducted in low(er) income countries, based on case scenarios or involving less than 100 patients were excluded.

**Review methods** Reviewers identified studies, extracted data and assessed the quality of the evidence independently and in duplicate. The Quality Assessment of studies of Diagnostic Accuracy included in Systematic Reviews -2 checklist was used to assess risk of bias. Raw data were extracted to create 2×2 tables and calculate sensitivity and specificity. ED patient volume and casemix severity of illness were investigated as determinants of triage systems' performance.

**Results** Sixty-six eligible studies evaluated 33 different triage systems. Comparisons were restricted to the three triage systems that had at least multiple evaluations using the same reference standard (Canadian Triage and Acuity Scale, Emergency Severity Index and Manchester Triage System). Overall, validity of each triage system to identify high and low-urgency patients was moderate to good, but performance was highly variable. In a subgroup analysis, no clear association was found between ED patient volume or casemix severity of illness and triage systems' performance.

**Conclusions** Established triage systems show a reasonable validity for the triage of patients at the ED, but performance varies considerably. Important research questions that remain are what determinants influence triage systems' performance and how the performance of existing triage systems can be improved.

## INTRODUCTION

Overcrowding of emergency departments (EDs) is a universal and ever-increasing problem.[1–3] Therefore, most EDs have a triage system in place to facilitate the prioritisation of patients. In recent years, several

### Strengths and limitations of this study

► The present study was based on a comprehensive and systematic search and attempted to include as many relevant studies as possible.
► This study is the first synthesis of the available evidence on triage systems as previous reviews have merely described the results of the individual studies.
► The majority of triage systems were evaluated by less than three studies. Even for the three most commonly used triage systems, few evaluations were available due to the variety of reference standards used.
► Due to the limited number of studies and the heterogeneity of study populations, it was difficult to compare results across studies and we could not provide summary estimates.

formal triage scales have been developed to standardise the approach to triage. These include among others the Australasian Triage Scale, the Canadian Triage and Acuity Scale (CTAS), the Emergency Severity Index (ESI) and the Manchester Triage System (MTS).[4–7]

It is important to evaluate the performance of triage systems for their ability to accurately distinguish between both high and low-urgency patients. The correct classification of high-urgency patients is related to patient safety, because misclassification of high-urgency patients to a low-urgency level causes delay in diagnosis and treatment, potentially leading to morbidity or mortality. The correct classification of low-urgency patients increases efficiency of the ED flow and reduces waiting times for the truly high-urgency ED visits.

Research regarding the performance of triage systems mainly consists of observational studies in heterogeneous populations using a variety of reference standards. Previous reviews have primarily described the results of these individual studies without combining and interpreting the evidence into overall conclusions.[8–10] A systematic appraisal of the

performance of commonly used triage systems can inform clinicians and policy-makers about the safety and efficacy of available triage systems and provide insights on which triage system is safe and efficient to use. Moreover, it can highlight gaps in current research and propose directions for future studies.

The aim of this systematic review is to provide a comprehensive overview of current evidence on the performance of triage systems. We assessed and compared the performance of the most commonly used triage systems for the prioritisation of high and low-urgency patients at the ED, compared with any reference standard that is a proxy of true patient urgency. Furthermore, we aimed to investigate whether patient volume and casemix at the ED are determinants of triage systems' performance.

## METHODS

### Search strategy

Meta-analysis of Observational Studies in Epidemiology guidelines were followed for the conduct of this study.[11] We conducted a systematic review using a broad search strategy to identify all studies assessing the performance of triage systems in emergency care when compared with any reference standard that is a proxy of true patient urgency. A search strategy was developed by a health sciences librarian and included Medical Subject Headings and text words related to triage, emergency care and validity (online supplementary appendix 1). We searched EMBASE, Medline OvidSP, Cochrane central, Web of science and CINAHL databases from 1980 to 2016 with the final update in December 2018.

### Study eligibility

Studies were selected that assess the performance of triage systems in emergency care with a defined outcome measure as a proxy of true patient-urgency. We selected studies based on the following PICO:

Population: We included studies evaluating triage in the unselected group of patients attending the ED. We excluded studies restricted to specific patient subgroups (such as patients with specific diseases).

Interventions: We included any studies assessing ED triage systems, defined as any tool aimed to classify patients at the ED based on the urgency or severity of their condition. We did not include studies evaluating trauma triage systems or early warning scores.

Comparators: Since no golden standard for the evaluation of triage systems exists, we included all studies evaluating the performance of triage systems using one or more defined reference standard as a proxy for true patient urgency.

Outcome: We defined outcome as the sensitivity and specificity of the triage system for the identification of high-urgent and low-urgent patients. A priori, we selected mortality at the ED and Intensive Care Unit (ICU) admission after the ED visit as reference standard for high-urgency and discharge home after the ED visit as the reference standard for low urgency. We additionally considered any other reference standard with sufficient evaluations.

Letters, abstracts, reviews, conference proceedings and case reports were excluded as well as studies not written in English. We excluded studies with less than 100 patients and studies based on case scenarios because these studies have of a high risk of bias. Moreover, we excluded studies conducted in low or lower income economies.[12] The unique characteristics of EDs in these countries, including the number of patients, epidemiology of diseases and available resources, make study results difficult to compare to middle or higher income countries. Two reviewers (MvV and NS or JMZ and VvdH) independently assessed eligibility for inclusion. Disagreements in article selection were resolved through discussion.

### Data extraction

Two reviewers (JMZ and VvdH) independently extracted data from each of the included studies. A single study could consist of multiple 'triage evaluations', defined as the analysis of a single triage system in a single age group. Predefined age groups were (1) children; (2) adults or a combination of age groups and (3) elderly. For each of the triage evaluations, the reviewers extracted the total number of included patients, the number of patients in each of the urgency categories of the triage system, the type of reference standard used and the number of patients with a positive reference standard in each urgency category. If studies were based on overlapping data, we used the results from the most recent publication. For descriptive purposes, we also collected data on study design and methods, patient demographics, and characteristics of the settings in which the study was performed.

### Quality assessment

Two reviewers independently assessed quality of the selected articles using the Quality Assessment of studies of Diagnostic Accuracy included in Systematic Reviews-2 (QUADAS-2) checklist.[13] The QUADAS-2 evaluates four domains: patient selection, index test, reference standard, and flow and timing. Each domain is assessed in terms of risk of bias, and the first three domains also in terms of applicability. Because triage systems have some specific features as compared with other diagnostic tests, we adjusted the 'reference standard domain' to make it applicable to our research question. We did not appraise whether the reference standard was interpreted without knowledge of the result of the index test, because this is unlikely when triage is applied in routine care. We did evaluate, however, whether data on the outcome were collected blinded to the result of the index test. Moreover, we did not judge the applicability of the reference standard because there is no consensus on this topic.[14] Therefore, we included all studies with reference standard

that were a proxy for patient urgency. Any disagreements between reviewers were resolved in a consensus meeting.

## Data analysis

We used descriptive analyses to provide an overview of the available evidence on triage systems. Further analyses were restricted to triage systems that underwent at least three evaluations with the same reference standard. The primary outcome of our review was sensitivity and specificity of each of the triage systems for the identification of high and low-urgency patients. Because there is no golden standard to determine 'true' patient urgency, we a priori selected three reference standards as proxy for patient urgency. We considered mortality at the ED and ICU admission after the ED visits as reference standard for high urgency, and discharge after the ED visit (ie, patients not admitted to hospital) as reference standard for low urgency. Although these measures are not perfect, they approximate the desired outcome: most patients who die at the ED or require ICU admission are of high urgency, while most patients who are discharged after the ED visit are not. Moreover, these measures are suitable for the analyses of large datasets and commonly reported in research on triage systems.[14] In addition, we considered any other reference standard with sufficient evaluations in the same triage system.

We calculated two by two tables of triage system against the reference standard for each individual study. Because triage systems are ordinal scales, we dichotomised the urgency categories into a high-urgency and low-urgency group. High urgency was defined as triage urgency level 1 (three-level systems) or triage urgency level 1 and 2 (four-level and five-level systems).

We calculated sensitivities and specificities, and presented the results as forest plots. We aimed to summarise the diagnostic accuracy data using a bivariate random effects model, but due to the substantial heterogeneity between studies, this was not possible. For clinical practice and for benchmarking purposes, we calculated the proportion of patients with a positive reference standard per urgency category. These results are displayed in a bar chart to enable comparison between studies and between triage systems.

We hypothesised that ED patient volume and casemix severity of illness were determinants of triage systems' performance. Therefore, we decided that if a sufficient number of studies were identified, we would investigate the effect of these determinants on triage systems' performance using subgroup analyses. We considered annual ED census as a marker of patient volume and the percentage of hospitalised patients as a marker of casemix severity of illness.

Computations were carried out with SPSS Statistics V.21.0 and figures were created using Review Manager V.5.3 or R V.3.2.0.[15–17]

## Patient and public involvement

Patients and public were not involved in this study.

## RESULTS

A total of 12 684 papers were identified in the electronic search, of which 66 were included in the final selection (figure 1).

The majority of studies were conducted in tertiary or university hospitals (n=46; 70%) and conducted in Europe/Central Asia or North America (n=45; 68%). Forty-nine (74%) were single-centre studies. A complete overview of the selected studies is presented in online supplementary appendix 2.

Forty-four studies (67%) had a high risk of bias in at least one domain, and 17 studies (26%) had a high risk of bias in two or more domains (figure 2 and online supplementary appendix 3). The most common causes of concern were application of multiple triage systems for the same patient by the same nurse or by multiple nurses without blinding, retrospective retrieval of reference standard information without blinding for the triage outcome, and substantial amounts of missing data. In 11 studies (17%), there were concerns regarding applicability.

## Triage systems

A total of 33 different triage systems were evaluated. The most commonly evaluated triage systems were the ESI (n=22), the MTS (n=15) and the CTAS (n=13). Other triage systems included the Taiwan Triage System (n=4), Australasian Triage Scale (n=3), South African Triage Scale (n=3), Netherlands Triage System (n=2) and Soterion Rapid Triage System (n=2). For 25 triage systems, only one evaluation was published. These included nine local or informally structured triage systems. The median sample size was 1496 in children (range: 510–550 940), 1447 in adults (range: 100–316 622) and 929 in elderly (range: 773–1903). In total, 89 individual triage evaluations were reported: 34 (38%) in children, 52 (58%) in adults, a combination of age groups or an unspecified population and 3 (3%) in elderly.

### Reference standards

A variety of reference standards were used and the majority of studies reported multiple reference standards (online supplementary appendix 4). Twelve studies used mortality at the ED as a reference standard, 13 studies ICU admission and 47 studies hospital admission. Other commonly reported reference standards were length of stay at the ED (27 studies), resource use at the ED according to the ESI criteria (14 studies), expert opinion (9 studies) and costs (8 studies). Because definitions of time to mortality, resource use and expert opinion were not consistent across studies, these results could not be compared. Moreover, length of stay at the ED and costs are outcome measures that are strongly dependent on ED characteristics, and could therefore not be used to make a comparison between studies.

## The most commonly evaluated triage systems

We will further restrict our analyses to the triage systems with at least three evaluations using the same reference

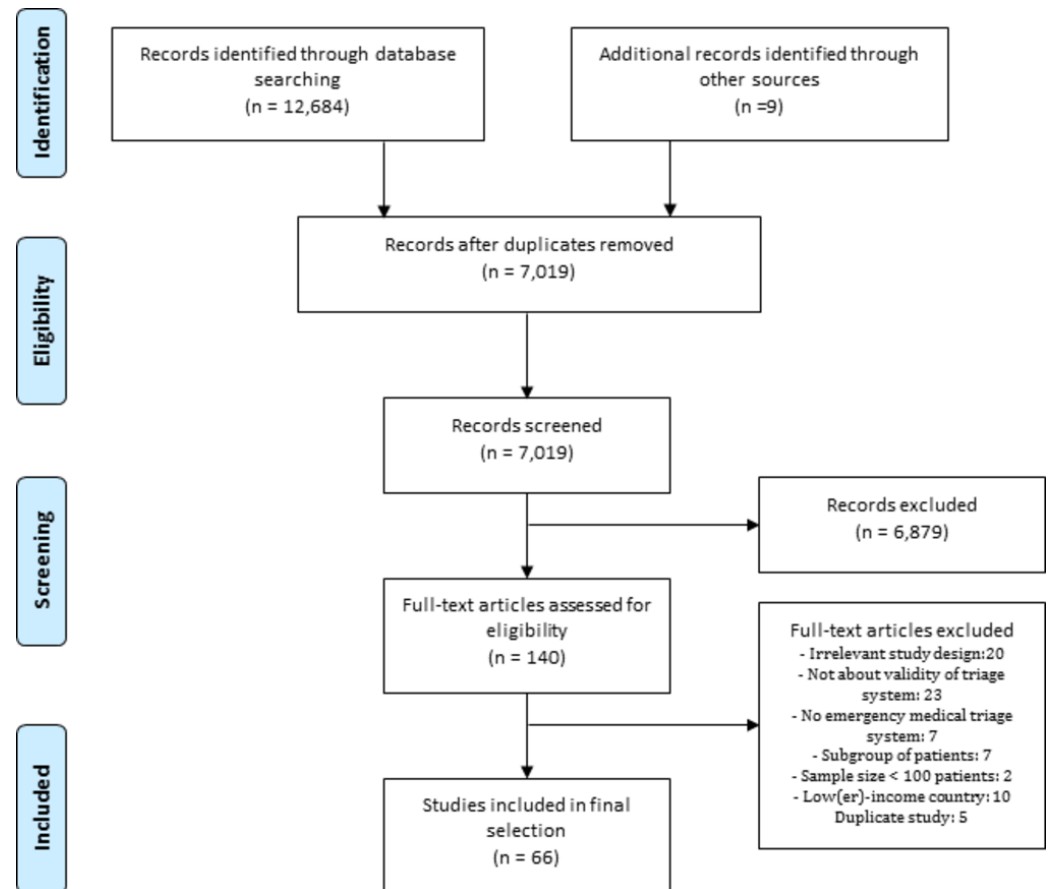

**Figure 1** Flow diagram of study selection process.

standard. This final selection includes studies evaluating the ESI, CTAS and MTS. Characteristics of these triage systems and a summary of the available evidence are presented in table 1. The ESI, CTAS and MTS were all evaluated in settings with a different patient volume as indicated by annual hospital census, and a different casemix as indicated by percentage of hospitalisation. For each of these triage systems, the majority of studies had a risk of bias in at least one domain.

### Accuracy of triage systems to identify high-urgent patients

Mortality at the ED was reported in seven evaluations of our final sample: five evaluations in adults and two in children. Because of this low number of studies and the very low reported mortality rates (on average 0.2% in adults and <0.01% in children), it was not possible to perform comparative analyses.

ICU admission was reported in five evaluations in adults (two ESI, three MTS) and four in children (three CTAS, one MTS). Overall, sensitivity for ICU admission was moderate to good, ranging from 0.58 (95% CI 0.48 to 0.68) to 0.88 (95% CI 0.70 to 0.96) in adults and 0.71 (95% CI 0.66 to 0.77) to 0.93 (95% CI 0.89 to 0.95) in children. A clear difference in performance between the triage systems was not visible (figure 3).

Regardless of the triage system used, most of the ICU admitted patients were allocated to one of the two highest triage categories (online supplementary appendix 5). The exact proportion of ICU admitted patients in each triage category was highly variable, even within studies evaluating the same triage system. For example, the proportion of ICU admitted adults in MTS category 1 ranged from 21% to 79%. The number

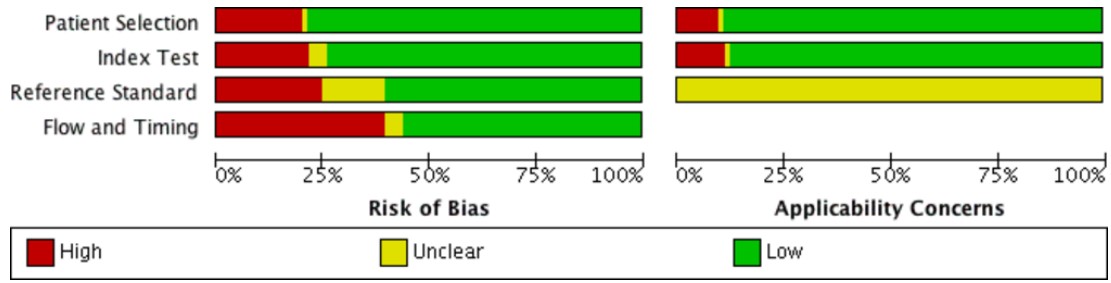

**Figure 2** Risk of bias of included studies.

**Table 1** Evidence summary of the most commonly used triage systems

| | Canadian Triage And Acuity Scale | Emergency Severity Index | Manchester Triage System |
|---|---|---|---|
| **Triage system characteristics** | | | |
| Description | List, based on presenting signs and symptoms | Flowchart, based on physical signs and expected resource use | Multiple flowcharts, based on presentational signs and symptoms |
| No of levels | 5 | 5 | 5 |
| Classification and waiting time | Level I, Immediate<br>Level II, 15 min<br>Level III, 30 min<br>Level IV, 60 min<br>Level V, 2 hours | Immediate<br>Emergent, 14 min<br>Urgent, 60 min<br>Semiurgent, 2 hours<br>Non-urgent, 24 hours | Immediate<br>Very urgent, 10 min<br>Urgent, 60 min<br>Standard, 2 hours<br>Non-urgent, 4 hours |
| **Quantity of evidence** | | | |
| Total no of evaluations | 13 | 21 | 15 |
| Evaluations in children | 9 | 4 | 7 |
| Evaluations in elderly | 1 | 2 | 0 |
| **Diversity of evidence (range)** | | | |
| No of hospitals per study | 1–12 | 1–7 | 1–4 |
| Inclusions per study | 481–550 940 | 180–37 974 | 872–31 622 |
| Hospital census | 10 000–75 000 | 10 000–90 000 | 7000–190 000 |
| Hospitalisation rate | 8%–47% | 10%–62% | 5%–33% |
| **Risk of bias** | | | |
| High risk of bias in at least one domain | 54% | 81% | 67% |
| High risk of bias in >1 domain | 15% | 23% | 13% |

of studies, however, was too small to assess whether this variability was present in all triage systems and whether this could be explained by study or setting related factors.

### Accuracy to identify low-urgent patients

Hospital admission or discharge after the ED visit was reported as a reference standard in 14 evaluations in adults and 15 in children. Overall, specificity of the triage systems to accurately classify patients discharged home as low urgent ranged from 0.64 (95% CI 0.62 to 0.66) to 0.98 (95% CI 0.95 to 0.99) in adults and 0.69 (95% CI 0.66 to 0.72) to 0.96 (95% CI 0.94 to 0.98) in children (figure 4). Again, sensitivities and specificities were highly variable within each of the triage systems. None of the triage systems showed a marked better specificity compared with the others.

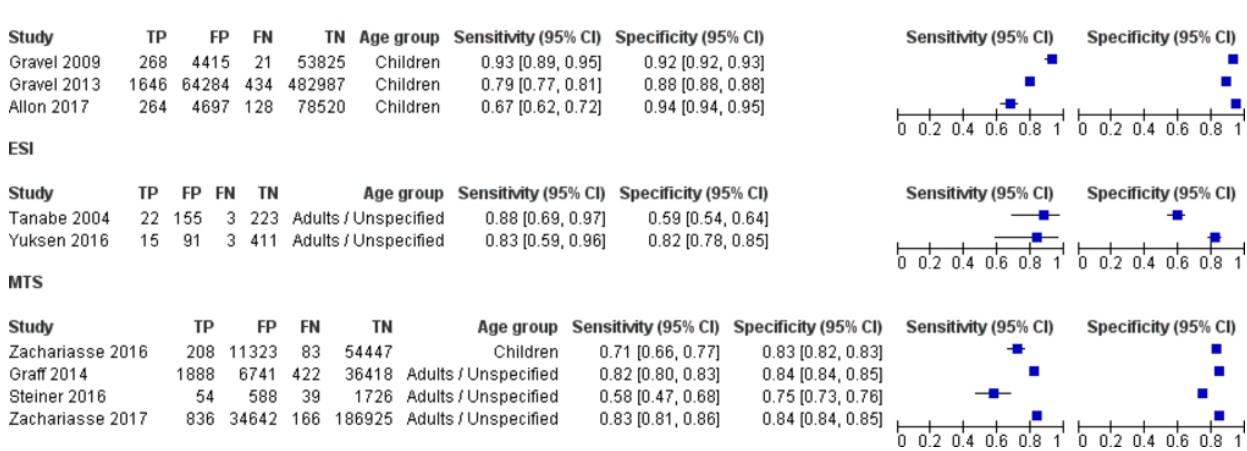

**Figure 3** Sensitivity and specificity of triage systems for identifying high-urgency patients as defined by ICU admission. CTAS, Canadian Triage and Acuity Scale; ESI, Emergency Severity Index; FP, false positive; FN, false negative; MTS, Manchester Triage System; TP, true positive; TN, true negative.

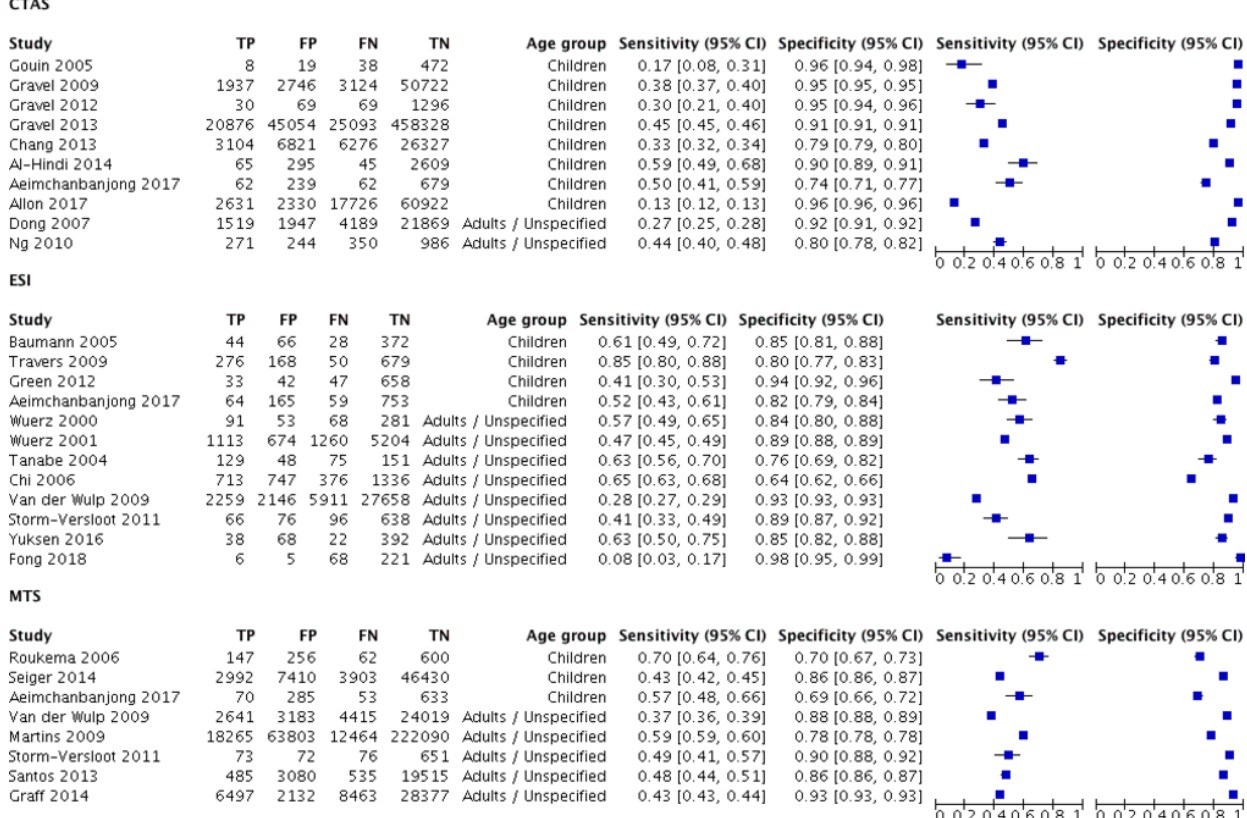

**Figure 4** Sensitivity and specificity of triage systems for identifying low-urgency patients as defined by discharge home after the ED visit. CTAS, Canadian Triage and Acuity Scale; ED, emergency department; ESI, Emergency Severity Index; FP, false positive; FN, false negative; MTS, Manchester Triage System; TP, true positive; TN, true negative.

The proportion of patients discharged after the ED visit increased from the higher to the lower urgency categories in all triage systems (online supplementary appendix 5). Again, there was a large variability within triage systems and substantial overlap between triage systems. In adults, the MTS seemed to have a higher variability compared with the other triage systems, but in children, variability was greater for the CTAS.

The only additional reference standard with sufficient evaluations was resource use according to the ESI criteria (online supplementary appendix 6).

### Direct comparison of triage systems

A total of 13 studies directly compared two or more triage systems. Most of these studies, however, were assessed as having a high risk of bias in the index test domain, because triage was performed by the same nurse or without blinding. Performing triage, while using different triage systems sequentially, is likely to reduce the differences between triage systems. Therefore, these results should be interpreted with caution.

In 10 studies, an established five-level triage system was compared with a local or informally structured triage system with three or four levels. Seven of these studies reported that the five-level triage system provided better discrimination or better sensitivities and specificities than the local triage system and should be preferred.[18–24] One study in children found that the local triage system performed better than the established triage system (CTAS).[25]

Two studies comparing the ESI with the MTS in adults found that sensitivities and specificities for hospital admission and the prediction of mortality were largely similar.[26 27] In one of these studies, the MTS undertriaged a smaller proportion of patients compared with the ESI (8.3% vs 13.5%) at the cost of a larger proportion of 'overtriage'.[26] One study in adults observed no statistically significant difference between the CTAS and ESI regarding the prediction of ED resource utilisation and immediate patient outcomes.[28] One study in children compared the ATS and ESI and found similar sensitivities and specificities for the identification of patients requiring hospital admission.[29] One single-centre study in children compared five triage systems (ATS, CTAS, ESI, MTS and a local triage system called the Ramathibodi Triage System) and concluded that the ESI showed the best validity for predicting hospital admission (Area Under the Curve 0.78, 95% CI 0.74 to 0.81).[30] In this study, the local triage system showed the highest sensitivity (50%) and the ATS the highest specificity (94%).

### Determinants of triage systems' performance

The number of studies per triage system was too small to perform subgroup analyses based on annual census or percentage hospitalisation. As an explorative analysis, we ordered all selected studies that used hospital admission

as a reference standard based on annual census, and percentage hospitalisation (online supplementary appendix 7). There was no clear association between patient volume or casemix and triage systems' sensitivity and specificity. A lower specificity for hospitals with the largest annual census and highest percentage hospitalisation could not be ruled out, but requires a larger number of studies.

## DISCUSSION

In a systematic review of 66 observational studies evaluating triage systems, we found that numerous different triage systems are being used but that many lack a rigorous evaluation. The most commonly used and evaluated triage systems, CTAS, ESI and MTS, show a moderate to good validity to identify high and low-urgency patients. Their performance, however, is highly variable and differences in study design, study populations and reference standards make a comparison of the available evidence difficult. Although based on a limited number of studies, no clear association between patient volume and casemix severity of illness could be found.

### Strengths and weaknesses

This is the first study that evaluates the performance of triage systems in a meta-analysis. Previous reviews have merely described the results of the individual studies without synthesising the evidence. Moreover, none of the published reviews looked at other factors that determine triage systems' performance, such as ED characteristics to compare evidence from different studies.[8–10 31–34]

Our review is based on a comprehensive search developed with a research librarian, includes duplicate assessment of eligibility and risk of bias, and duplicate data abstraction. Furthermore, the research question is based on a relevant and practical clinical issue. Triage systems are used worldwide to prioritise patients in the ED, but robust evidence on their performance is lacking.

The results of this review, however, should be interpreted taking into consideration the limitations of the underlying evidence. We included 66 studies in our review, but the majority of the 33 triage systems were evaluated by only one study. Therefore, we could only evaluate the three most frequent used triage systems: CTAS, ESI and MTS. Even for these commonly used triage systems, few evaluations were available due to the variety of reference standards. Although the triage evaluations included the whole age spectrum, studies targeted at elderly patients were scarce. It is important to evaluate triage systems' performance separately for the most vulnerable populations at the ED, specifically children and elderly. In these patient groups, the spectrum of disease, the presence of non-specific signs and symptoms and progression of disease course differ from that in adult patients.

We are not aware of any (randomised) controlled trial that investigates the effect of triage on patient outcome. Therefore, we conducted a systematic review of observational studies. Comparing observational studies is challenging because the effect of a triage system cannot be assessed independently of its context. Likely, other factors such as ED and hospital characteristics or local practices and training have influenced the results of the included studies. We chose to display the results of different studies in a forest plot. Because of the limited number of studies and heterogeneity of the study populations, it is difficult to compare the results from different triage systems and these plots should be interpreted with caution.

We aimed to explore heterogeneity between studies and more specifically the effect of differences in patient load and casemix severity of illness. Unfortunately, due to the small number of studies per triage systems, we could not draw strong conclusions about the relation between these factors and triage systems' performance. There are more potential factors that could affect triage systems' validity, such as the local infrastructure, the experience and training of the triage nurse, the presence of a computerised triage application or variations in disease epidemiology. Moreover, since most included studies had a risk of bias in at least one domain, we cannot rule out that study design and methodological quality have led to heterogeneity of the results as well.

We predefined mortality at the ED and ICU admission as reference standards for high patient urgency and discharge home as a reference standard for low patient urgency. Due to the relatively low number of studies reporting mortality and the low mortality rate of patients in the ED, we could not use it as a reference standard. ICU and hospital admission are feasible reference standards for large study populations, and theoretically, criteria for ICU and hospital admission should be reasonably comparable between settings. It is possible, however, that ED and hospital characteristics and local practices result in differences in the decision to admit a patient between EDs.

We restricted our review to triage systems in high-income and higher-middle-income countries. We applied this selection because EDs in lower income countries have their own unique characteristics and challenges. Several recently published studies have addressed triage in low-income settings.[35 36]

### Implications and future research

Our review identified 33 triage systems for which at least one evaluation was published. Probably, there are more triage systems in use, which are not formally evaluated. There are several advantages, not addressed in this review, of using an established triage system over a local triage scale. Beside that performance of the most commonly used triage systems is known, they have a formal governance structure and undergo regular updates. Moreover, there are standardised implementation guidelines and training programmes available.[37–39] The CTAS, ESI and MTS all show a reasonable performance for triage at the ED. Our results do not suggest that one of the established triage systems should be preferred over the other.

Our review indicates that large variation of performance exists even in studies assessing the same triage system. This suggests that other factors influence triage systems' performance. Consequently, generalisability of individual studies evaluating a triage system is low and a study on triage validity in one setting may not apply to a setting with different characteristics or in a different healthcare system. Our review demonstrates that the majority of studies evaluating triage systems were conducted in a single centre. Furthermore, most multicentre studies provided only pooled results. Yet, multicentre studies using similar study designs and reference standard definitions are needed to evaluate the range of triage systems' performance in different settings. More importantly, these studies can provide valuable insights in determinants of triage systems' performance and areas of improvement.

The extensive use of triage systems in clinical practice contrasts with the limited number of studies evaluating their performance. Triage systems were typically developed based on expert opinion and implemented out of clinical necessity. They are mostly used in their country or region of origin: the MTS is widely used in the UK and Europe, the ESI in the USA, and the CTAS in Canada and in French-speaking countries.[40] Since most EDs already have experience with a certain triage system and some triage systems are recommended by national guidelines, it could be worthwhile shifting away from the focus on triage systems' performance towards the improvement of the established triage systems. Our review suggests that there is room for improvement of all triage systems regarding both the correct identification of high-urgency and low-urgency patients.

The most commonly used triage systems, CTAS, ESI and MTS, have a reasonable validity for the triage of patients at the ED. Important research questions that remain are what determinants influence a triage systems' performance and how the performance of existing triage systems can be improved.

**Correction notice** This article has been corrected since it first published online. The open access licence type has been amended.

**Acknowledgements** We thank Wichor Bramer, biomedical information specialist in Erasmus University Medical Centre, Rotterdam, for his help with developing the search strategy

**Contributors** JMZ and VvdH contributed equally to this work. All authors substantially contributed to the conception and design of the study and interpretation of the findings. JMZ, KM-J and HAM conceived the study idea. JMZ, VvdH, NS and MvV conducted the systematic review including screening of abstracts and full text and risk of bias assessment. JMZ and VvdH extracted data, performed the analyses and wrote the first draft of the manuscript. All authors revised it critically for important intellectual content and gave their approval of the final version. All authors had full access to all of the data (including statistical reports and tables) in the study and can take responsibility for the integrity of the data and the accuracy of the data analysis. HAM is guarantor.

**Funding** The authors have not declared a specific grant for this research from any funding agency in the public, commercial or not-for-profit sectors.

**Competing interests** KM-J is chair of the Manchester Triage Group.

**Patient consent for publication** Not required.

**Provenance and peer review** Not commissioned; externally peer reviewed.

**Data sharing statement** No additional data are available.

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
