## [Reviewer comments · BMJ Open]

ARTICLE DETAILS

TITLE (PROVISIONAL)	THE PERFORMANCE OF TRIAGE SYSTEMS IN EMERGENCY CARE: A SYSTEMATIC REVIEW AND META-ANALYSIS
AUTHORS	Zachariasse, Joany; van der Hagen, Vera; Seiger, Nienke; Mackway-Jones, Kevin; van Veen, Mirjam; Moll, Henriette

VERSION 1 - REVIEW

REVIEWER	Bhakti Hansoti Johns Hopkins, USA
REVIEW RETURNED	14-Oct-2018

GENERAL COMMENTS	Overall this is a well written paper that is able to clearly and concisely articulate the purpose, methodology and key findings. The difficulties of this analysis and the overall results are easy to follow. My only concerning is that several studies seem to be missing. I would ask the authors to compare the included studies to that of the recent s/r by Hinson and Scott Levine from Johns Hopkins soon to be published in the annals of emergency medicine. From the overview of this reviewer all the South African Triage Score (SATS) written by Dalwai and Twomey seem to be missing and this raises concerns for other omissions. I would request that authors to provide a detailed account of why study were excluded to be able to account for these omissions.
--

REVIEWER	lee Wallis University of Cape Town South Africa
REVIEW RETURNED	24-Oct-2018

GENERAL COMMENTS	Thank you. Generally, this is well done, with robust searching and quality assessment methods. However, despite the solid article identification and interpretation methods I have serious concerns. I note that BMJ uses UK English. The use of US English suggests that has been copied across from a submission to another journal, without much attention to the BMJ's requirements. The use of the term "western countries" is inappropriate (especially since authors have included studies from all over the world). Using it as an inclusion criterion is not well defended. It seems like the objective was to see which of a narrow range of triage tools worked in a very selective setting of high-income
--

	countries. If the intention was that, state it more clearly and be very clear why you are excluding studies in low resource settings. In that case, the background and discussion need to be very focussed on the fact that this is a review of 3 tools for high income settings. Authors have missed key literature, including 2 systematic reviews in the last year - one looking at adult tools in low resource settings and one looking at tools for children in the same setting. Why have they been excluded from the background and discussion? Why are their results not applicable here? The authors exclude studies using cases, despite the validity of such methodology, and then use mortality as the outcome marker, despite the fact that triage is not intended to detect mortality and, in fact, there are several better proxy markers than this. Similarly, there are multiple confounders in the decision to admit or discharge, so the use of discharge as a proxy has significant limitations. The reference list appears twice.
--	--

VERSION 1 – AUTHOR RESPONSE

REVIEWERS' COMMENTS TO AUTHOR:

Reviewer: 1

Reviewer Name: Bhakti Hansoti

Institution and Country: Johns Hopkins, USA Please state any competing interests or state 'None declared': None declared

Overall this is a well written paper that is able to clearly and concisely articulate the purpose, methodology and key findings. The difficulties of this analysis and the overall results are easy to follow. My only concerning is that several studies seem to be missing. I would ask the authors to compare the included studies to that of the recent s/r by Hinson and Scott Levine from Johns Hopkins soon to be published in the annals of emergency medicine. From the overview of this reviewer all the South African Triage Score (SATS) written by Dalwai and Twomey seem to be missing and this raises concerns for other omissions. I would request that authors to provide a detailed account of why study were excluded to be able to account for these omissions.

We thank the reviewer for her review of our paper and for mentioning the recent publication by Hinson et al.

We critically reviewed the reference list of this systematic review and we identified 30 papers that were not included in our review. All of them fulfilled one of our exclusion criteria: 15 were not related to validity of the triage system (these were mainly reliability studies), 12 were based on specific subgroups of patients such as patients with myocardial infarction or sepsis, and 3 were not written in English.

In our review, we did not intentionally exclude articles about the South African Triage Score (SATS). We have included one article by Twomey about the SATS and our updated search update included one more article about the SATS by Meyer et al.

To be specific about our reasons to exclude other papers by Dalwai and Tomey: the study by Dalwai et al, 2014 was identified through our search but excluded because it was based on case scenarios

and conducted in Pakistan (defined as low/lower middle income country); the study by Twomey et al, 2011 was identified through our search but excluded because it only evaluated reliability and not validity.

We also provide a more detailed explanation about our exclusion criteria in the response to the second reviewer.

Reviewer: 2

Reviewer Name: Lee Wallis

Institution and Country: University of Cape Town, South Africa Please state any competing interests or state 'None declared': none declared

Thank you. Generally, this is well done, with robust searching and quality assessment methods.

We thank the reviewer for his review of our paper.

However, despite the solid article identification and interpretation methods I have serious concerns. I note that BMJ uses UK English. The use of US English suggests that has been copied across from a submission to another journal, without much attention to the BMJ's requirements.

We would like to clarify to the reviewer that we carefully followed the BMJ Open author requirements. BMJ open does not mention any specific language requirements and The BMJ house style allows both English and American spelling (<https://www.bmj.com/about-bmj/resources-authors/house-style>).

We agree, however, with the author that the use of American English may be distracting, and therefore we changed to manuscript to British English spelling.

The use of the term "western countries" is inappropriate (especially since authors have included studies from all over the world).

We agree with the author that "Western countries" is indeed an erroneous term and we changed it to "high or higher middle income countries".

Using it as an inclusion criterion is not well defended. It seems like the objective was to see which of a narrow range of triage tools worked in a very selective setting of high-income countries. If the intention was that, state it more clearly and be very clear why you are excluding studies in low resource settings. In that case, the background and discussion need to be very focussed on the fact that this is a review of 3 tools for high income settings.

We thank the reviewer for addressing this issue and agree that the exclusion of low and lower income countries need a more detailed explanation.

First and foremost, we would like to emphasise the importance of research on triage in low- and lower-middle income countries. We believe emergency departments in these countries have unique characteristics, such as the number of attending patients, epidemiology of diseases, and available resources that are different from those in higher (middle) income countries. This is also reflected by the development of specific triage systems targeted at these countries, such as the WHO Emergency Triage Assessment and Treatment (ETAT) guidelines.

Writing the review protocol, we discovered enormous heterogeneity between studies and study populations. In order to draw any meaningful conclusions we had to make a clear selection of patients and study settings that would allow us to combine the results of studies into a (meta-)analysis. Since

our review is already very elaborate we decided to leave out this specific subgroup of settings to improve readability of our manuscript.

We acknowledge that country income level may not be the ideal criterion to classify emergency departments. We chose the World Bank classification because it is an objective and independent selection. We would like to add that the World Bank criteria classify 137 (63%) of all 218 countries as high or upper middle and our selection criteria therefore still include the majority of countries worldwide.

We have extended the statement on the exclusion of low and lower income countries in the Methods section: "Moreover, we excluded studies conducted in low or lower-income economies because their unique characteristics, including the number of patients, epidemiology of diseases and available resources, make study results difficult to compare to most emergency departments in middle or higher income countries."

Moreover we added a section in the limitations section of the Discussion

"We restricted our review to triage systems in high and higher-middle income countries. We applied this selection because EDs in lower income countries have their own unique characteristics and challenges. Several recently published studies have addressed triage in low-income setting."

Authors have missed key literature, including 2 systematic reviews in the last year - one looking at adult tools in low resource settings and one looking at tools for children in the same setting. Why have they been excluded from the background and discussion? Why are their results not applicable here?

We thank the reviewer for mentioning these recently published reviews. Since our study focusses on triage in high and middle income countries we have not mentioned these two studies in our review.

We added a statement in the Discussion section (mentioned above) that refers to systematic review in low-income countries.

The authors exclude studies using cases, despite the validity of such methodology,

We agree with the author that case scenarios can provide important insights in the validity of triage systems. However, paper scenarios already involve all information that is needed for a complete triage assessment. Nurses do not have to ask the patient for specific information and do not have to interpret patient signs and symptoms. This is an important but also very complex aspect of the triage, but not assessed in paper scenarios. Therefore we do not think results of these studies cannot be compared to the triage of "real patients". Besides this, most studies using case scenarios also involved less than 100 cases, which is also an exclusion criterion for our study.

and then use mortality as the outcome marker, despite the fact that triage is not intended to detect mortality and, in fact, there are several better proxy markers than this. Similarly, there are multiple confounders in the decision to admit or discharge, so the use of discharge as a proxy has significant limitations.

We agree with the reviewer that triage is not intended to detect mortality. We propose in our manuscript that the main objective of triage is to detect patient urgency and we attempted to use the best proxy for this outcome. We acknowledge that mortality and discharge as proxy for respectively high and low patient urgency are far from perfect and have their own limitations. That is why we also choose to include studies using other reference standards as well. We encountered multiple other reference standards that were used in the included studies. Unfortunately, only resource (according to ESI criteria) was used in a sufficient number of studies to be included in the review.

The reference list appears twice.

We thank the reviewer for addressing this issue. We included two reference lists: one for the manuscript and one for the supplementary material. The latter also includes the references of the studies that were included in the review.

VERSION 2 – REVIEW

REVIEWER	lee Wallis University of Cape Town South Africa
REVIEW RETURNED	22-Jan-2019

GENERAL COMMENTS	much improved,thank you
-------------------------